# Prader–Willi Syndrome and Hypogonadism: A Review Article

**DOI:** 10.3390/ijms22052705

**Published:** 2021-03-08

**Authors:** Cees Noordam, Charlotte Höybye, Urs Eiholzer

**Affiliations:** 1Centre for Paediatric Endocrinology Zurich (PEZZ), 8006 Zurich, Switzerland; urs.eiholzer@pezz.ch; 2Department of Pediatrics, Radboud University Medical Centre, 6525 GA Nijmegen, The Netherlands; 3Department of Endocrinology, Karolinska University Hospital, 111 52 Stockholm, Sweden; charlotte.hoybye@sll.se; 4Department of Molecular Medicine and Surgery, Karolinska Institute, 171 76 Stockholm, Sweden

**Keywords:** Prader-Willi syndrome, hypogonadism, child, adult, review, diagnosis, treatment, substitution

## Abstract

Prader-Labhart-Willi syndrome (PWS) is a rare genetic disorder characterized by intellectual disability, behavioural problems, hypothalamic dysfunction and specific dysmorphisms. Hypothalamic dysfunction causes dysregulation of energy balance and endocrine deficiencies, including hypogonadism. Although hypogonadism is prevalent in males and females with PWS, knowledge about this condition is limited. In this review, we outline the current knowledge on the clinical, biochemical, genetic and histological features of hypogonadism in PWS and its treatment. This was based on current literature and the proceedings and outcomes of the International PWS annual conference held in November 2019. We also present our expert opinion regarding the diagnosis, treatment, care and counselling of children and adults with PWS-associated hypogonadism. Finally, we highlight additional areas of interest related to this topic and make recommendations for future studies.

## 1. Introduction

Prader-Labhart-Willi syndrome (PWS) is a rare genetic disorder resulting from a lack of expression of the paternally derived chromosome, 15q11-q13. The clinical entity was described for the first time by Andrea Prader, Alexis Labhart and Heinrich Willi in 1956 and was called Prader-Labhart-Willi syndrome. Prader being a pediatric endocrinologist, Willi a gynaecologist and Labhart an adult endocrinologist-together overseeing the three principal periods of life with different clinical presentations of people with PWS. Over time the name of Labhart disappeared, apparently while in the United States the name was judged as too long. The middle name disappeared while it seemed less important and some people thought Willi was the first name of Professor Andrea Prader (personal communication Prader-Eiholzer). PWS is characterised by intellectual disability, behavioural problems and a hypothalamic dysfunction combined with specific dysmorphisms [1,2,3]. Hypothalamic dysfunction causes growth hormone deficiency, hypogonadism, and a dysregulation of energy balance associated with hypoactivity and insatiable hunger that results in increased fat and decreased muscle mass. In addition to controlling nutrient intake and enhancing physical activity, the beneficial effects of early treatment of growth hormone deficiency in PWS have been demonstrated: short stature is normalised and muscle mass enhanced [4,5]. Hypogonadism is distinct in males because sexual maturation stops at the early to mid-puberty stages [6]. This condition is less apparent in most girls, since an almost complete maturation and breast development occurs. However, spontaneous menarche is rather unusual [6]. While growth hormone therapy has been broadly accepted as a standard treatment in PWS, supplementing gonadal hormones is a matter of debate since the early 90s, mainly because of the presumed associated behavioural changes. Nevertheless, supplementing gonadal hormones is thought to be effective and beneficial for improving health and quality of life in PWS [7]. As part of the International Prader-Willi Syndrome Organisation annual conference held in Cuba in November 2019, a workshop dedicated to share experiences on gonadal function, puberty, sexuality and fertility in PWS confirmed the need for more information. Consequently, this paper aims to describe the current knowledge on hypogonadism in PWS children and adults with respect to diagnosis and treatment, based on current literature and the outcomes of this meeting. We present our expert opinion on the treatment, care and counselling of children and adults with PWS-associated hypogonadism.

## 2. Nature and Extent of Hypogonadism in PWS

### 2.1. Normal Sexual Development

Normal sexual development is the result of genetic factors, sexual hormones and germ cell production. Puberty involves the growth and maturation of gonads (testes and ovaries) and the appearance of sexual characteristics (e.g., sexual hair and breast development). In general, breast development in girls and testicular enlargement in boys (>3 mL) are the first signs of normal pubertal development. During puberty, menstruation and the onset of sperm emission occurs. The changes in puberty are caused by increased gonadal sex steroid secretion, primarily oestrogen in females and testosterone in males. Increased gonadal activity occurs because of increased stimulation by the pituitary gonadotropin luteinising hormone (LH) and follicle stimulating hormone (FSH).

Hypogonadism is the condition in which the production of sex hormones and germ cells are inadequate because of primary gonadal failure (primary hypogonadism) or hypothalamic pituitary dysfunction (secondary hypogonadism). Hypogonadism can be evident either at birth (e.g., indicated by clitoral hypoplasia in girls and micropenis and cryptorchidism in boys) or later in life (e.g., failure to enter puberty or complete sexual maturation).

### 2.2. Sexual Development in PWS

#### 2.2.1. Clinical Aspects

At birth, the prevalence of cryptorchidism and scrotal hypoplasia ranges from 75% to 95% [6,8,9,10,11]. Micropenis or small penis is reported to occur in males at rates of between 78% to 90%. In girls, the respective rates of clitoris and labia minora hypoplasia are 93% and 88% [9,11]. In contrast, Hirsch et al. reported normal penile length in infancy and early childhood, but small penile size in adulthood [6]. In the original description, Prader, Labhardt and Willi stated that puberty was delayed and incomplete [1]. The age of onset for boys is typically normal, but tends to show pubertal arrest at a mean chronological age of 13.5 years with a mean testicular volume of 4 mL and decelerating growth velocity; pubic hair growth reaches Tanner Stages 3–5 at this time due to premature pubarche [7].

In girls, the onset of puberty with breast development occurs at a normal age [6,12], although precocious puberty has been presented in several case reports [6,13,14]. While further development is delayed, Tanner Stage 5 development is usually noted past the normal age of 15.5 years [12]. Pubic hair developed early (i.e., Tanner Stages 4–5) in all girls studied [6]. Vaginal bleeding generally occurs between 13 and 34 years (mean age, 20) [6], although Siemensma et al. reported a mean age of 14.9 years at menarche for a sample of five girls (age range, 12.7–16.7) [12]. After menarche, regular bleeding are rare and oligomenorrhea is most prevalent. Some have bleedings, most likely caused by the unopposed action of oestrogens originating from adipose tissue [6].

Until now, there are no reports on PWS men having fathered a child, but evidence exists of at least seven pregnancies in women with PWS [15].

#### 2.2.2. Biochemical Aspects

In 1956, Prader and colleagues first reported low urinary 17-ketosteroid excretion in older males and concluded that testosterone levels were low [1]; they also measured increased gonadotropin secretion. Further work established a normal minipuberty in boys, associated with normally elevated LH, FSH and testosterone levels within the first few months of life [7,16]. Until puberty, inhibin B, LH and FSH levels remain at the lower end of the normal concentration ranges. At the onset of puberty, testosterone levels increase but remain low. Both LH and FSH also increase: LH levels usually only reach the low regions of the normal range and FSH levels are normal to high. Conversely, inhibin B levels decline to low and even undetectable levels in adulthood. This pattern is, together with the low anti-Müllerian hormone (AMH) serum levels, indicative of a primary testicular defect. The fact that LH and FSH levels do not increase to high levels could be interpreted as partial hypogonadotropic hypogonadism [6,7,8,17,18,19].

Siemensma et al. [12] reported on girls with PWS from 6 months until 22 years. AMH was normal, indicating the presence of primordial follicles. LH and oestrogen levels were normal before the age of 10 years. Thereafter, LH and FSH increased within the normal range, although LH was relatively low considering the low oestrogen levels and FSH relatively high. After 15 years, oestrogen levels were lower than in normal girls, LH levels were normal, and FSH slightly elevated. Inhibin B levels fell in the lower region of the normal range and inhibin A levels low to normal; small antral follicles were normally present, but with signs of impaired follicle maturation. AMH concentrations varied between normal to lower levels of the normal range. The authors interpreted their results as partial hypogonadotropic hypogonadism. Gross-Tsur et al. [16] outlined several patterns of hypogonadism in 16 adolescent and adult females (aged between 16 and 34 years) with varying combinations of primary (hypergonadotropic) and central (hypogonadotropic) hypogonadism. Concentrations of FSH and inhibin B were the most useful criteria to characterize the type of hypogonadism. Hirsch [6] added to the findings in adult women that inhibin B varied over time and was low or undetectable in the majority. Nonetheless, normal levels can be found, indicating potential fertility; repeat measurements are therefore recommended for contraceptive measures.

The type of hypogonadism-hypogonadotropic or hypergonadotropic or a combination of these-becomes apparent only in late adolescence or early adulthood and appears to stabilise after the age of 20 years [6,8].

#### 2.2.3. The Role of Orchidopexy

Eiholzer et al. [7] found that levels of gonadotrophins were not significantly different in boys with or without cryptorchidism until the age of 18 months. Inhibin B levels were significantly decreased after infancy. A specific gene expressed in testes (C15orf2) that lies on the PWS region of chromosome 15 might be involved in normal spermatogenesis and therefore explain the infertility and low inhibin B levels in boys and adult males with PWS [20]. There is no relation between age at orchidopexy and inhibin B levels, which suggests that cryptorchidism does not play a major role in hypergonadotropic hypogonadism.

#### 2.2.4. Interaction

The question has been raised as to whether the growth hormone deficiency present in PWS contributes to impaired gonadal function. Growth hormone substitution was shown to potentiate the Leydig cell response to human chorionic gonadotropin (hCG) in patients with hypopituitarism or hypogonadotropic hypogonadism [21,22]. Nevertheless, an arrest of gonadal development is still observed in all boys at pubertal age, despite otherwise efficacious growth hormone treatment [23].

The cause of cryptorchidism is not well established. Minipuberty is normal in PWS and associated with normal gonadotrophin and testosterone levels, and in utero testosterone levels are normal. Insulin-like factor-3 (INSL-3) protein is also adequately expressed in testes in PWS. Therefore, other factors have been suggested, for example that diminished abdominal pressure could hamper testicular descent, such as that observed in prune belly syndrome [7,17].

A candidate gene to the cause of hypothalamic hypogonadism, could be the necdin gene located on the human chromosome 15q and its chromosome 7-equivalent in mice. Loss of gene function in mice might result in impaired development of gonadotropin-releasing hormone neurons. Expression in the ovary is then important for factors involving oocyte development [24]. Another candidate is the melanoma antigen L2 (MAGEL2) gene and its lack of expression, which results in less ovulation in mice with smaller litter sizes-this gene is expressed in the hypothalamus [25].

Regarding the rare event of true precocious puberty observed in PWS, an imprinted gene coding for MKRN3 (makorin RING-finger protein 3) has been identified and is located in the critical region for PWS on chromosome 15 [26].

#### 2.2.5. Histological Aspects

There are few reports on gonadal histology in PWS. Vogels et al. [27] reported decreased and absent spermatogonia in testicular biopsies from eight prepubertal boys with PWS and one adult with only Sertoli cells. Katcher et al. [28] reported the absence of spermatogonia in PWS and also in males with well-descended testes. Matsuyama et al. [29] performed testicular biopsies during orchidopexy for cryptorchidism; spermatogonia was classified in nine boys (mean age, 2.2 years) and showed favourable histology in only two samples (Nistal I) and seven with unfavourable histology (Nistal scores II and III) where three had Sertoli cell-only syndrome. In all adults, LH and FSH levels were elevated and testosterone low.

After the onset of puberty when Sertoli cells mature, inhibin B production becomes dependent on normal spermatogenesis. Thus, inhibin B decreases when spermatogenesis does not take place.

There are two case reports on females with PWS: (1) an autopsy of a 21-year-old woman showed the lack of ovarian follicle development [30], whereas (2) examination of an ovary obtained from a 32-year-old woman during a caesarean birth revealed normal follicles in all stages of development [15].

### 2.3. Treatment

#### 2.3.1. Supplementing Gonadal Hormones in Normal Male and Female Hypogonadism

We searched PubMed with emphasis on publications from the last years using the keywords: “hypogonadism”, “treatment”, “substitution” in combination with the “similar articles” and “cited by” tools to retrieve current treatment guidelines as basis for the next section. In primary hypogonadism (or hypergonadotropic hypogonadism), sex hormone substitution with oestrogens or testosterone is considered the mode of treatment. For the treatment of central hypogonadism (i.e., hypogonadotropic hypogonadism), gonadotropic factors (LH and FSH) offer the potential to improve fertility in cases where this aspect is the primary therapeutic aim. Controlled trials on pubertal induction in normal male and female hypogonadism are virtually absent and current practice is based on expert opinion. The aim is to induce secondary sexual characteristics, increase muscle mass, establish peak bone mass and maximise final (adult) height. After induction, long-term sex hormone substitution is required to maintain normal serum sex hormone levels, sexual function, bone density and general well-being.

Boys and men. Most knowledge gained on the initiation and progression of pubertal development has been made with the application of intermediate-acting esters of testosterone (testosterone enanthate and testosterone cypionate) or a mixture of very short and short-acting esters (Sustanon^®^) [31,32]. A key case study from the group of Prader described the effects of puberty induction with androgens in 21 boys with bilateral anorchia, including a discordant pair of identical twins [33]. Long-acting testosterone enanthate (100 mg/month) was administered at a bone age of 12.8 years and continued with a higher dose of 250 mg/month by a bone age of 14.5 years; this regimen resulted in normal growth and pubertal progression in the affected twin, which was comparable to his normal twin brother. Most guidelines recommend an initial monthly intramuscular testosterone ester dose of 50 to 100 mg with gradual increases in dose over the course of 18–24 months for the induction of puberty in boys [34]; alternatively transdermal testosterone (10 mg) or oral testosterone undecanoate (40 mg). This generally recommended procedure induces secondary sexual characteristics and maximises final (adult) height [35]. In adults, dosages are individualised based on serum testosterone levels and well-being. A typical intramuscular testosterone ester dose is 250 mg every 3 weeks or 1000 mg every 10–14 weeks. Evaluation of mood is often recommended as changes can occur. Therapeutic levels for testosterone are reported to range from 450 to 600 ng/dL. In obese men, sex hormone-binding globulin (SHBG) and resultant testosterone levels can be low. Therefore, it is recommended to also measure free testosterone levels and SHBG in order to decide whether free testosterone should be measured to monitor and tailor therapy [36]. Haemoglobin levels should also be monitored as there is a risk of erythrocytosis when testosterone levels are too high. To monitor the effect of testosterone on the prostate, regular prostate specific antigen measurements are recommended after the age of 40 years.

Boys and men. Most knowledge gained on the initiation and progression of pubertal development has been made with the application of intermediate-acting esters of testosterone (testosterone enanthate and testosterone cypionate) or a mixture of very short and short-acting esters (Sustanon^®^) [31,32]. A key case study from the group of Prader described the effects of puberty induction with androgens in 21 boys with bilateral anorchia, including a discordant pair of identical twins [33]. Long-acting testosterone enanthate (100 mg/month) was administered at a bone age of 12.8 years and continued with a higher dose of 250 mg/month by a bone age of 14.5 years; this regimen resulted in normal growth and pubertal progression in the affected twin, which was comparable to his normal twin brother. Most guidelines recommend an initial monthly intramuscular testosterone ester dose of 50 to 100 mg with gradual increases in dose over the course of 18–24 months for the induction of puberty in boys [34]; alternatively transdermal testosterone (10 mg) or oral testosterone undecanoate (40 mg). This generally recommended procedure induces secondary sexual characteristics and maximises final (adult) height [35]. In adults, dosages are individualised based on serum testosterone levels and well-being. A typical intramuscular testosterone ester dose is 250 mg every 3 weeks or 1000 mg every 10–14 weeks. Evaluation of mood is often recommended as changes can occur. Therapeutic levels for testosterone are reported to range from 450 to 600 ng/dL. In obese men, sex hormone-binding globulin (SHBG) and resultant testosterone levels can be low. Therefore, it is recommended to also measure free testosterone levels and SHBG in order to decide whether free testosterone should be measured to monitor and tailor therapy [36]. Haemoglobin levels should also be monitored as there is a risk of erythrocytosis when testosterone levels are too high. To monitor the effect of testosterone on the prostate, regular prostate specific antigen measurements are recommended after the age of 40 years.

Girls and women. Most evidence on the induction of puberty with oestrogens in girls has been gathered in Turner syndrome. While it is more common to begin therapy from 12 years of age, the earliest time to treat is reported at 10 years-more recent studies point to the advantages of an earlier onset of therapy in terms of final height, cognitive development and uterine maturation [37,38]. Nevertheless, the appropriate age to start hormone substitution remains a matter of debate. Consensus only exists on the initial application of low-dose oestrogens (i.e., one-eighth to one-quarter of the recommended adult dose), which can be gradually increased at intervals of 6–12 months. Monitoring focuses either on the clinical effect (Tanner stage, bone age and uterine growth) or by applying the ultrasensitive oestradiol assay. The route of administration is oral or preferably transdermal to eliminate the first pass effect in the liver. The use of low dose oestradiol formulations (e.g., 0.1 mg daily) is important. Transdermal oestradiol administration begins at low doses (0.05–0.07 μg/kg nocturnally). In older girls, the starting dose can range from 0.08 to 0.12 μg/kg and is slowly increased over 12–24 months, after which time cyclic gestagen is added (or after the first menstrual bleed) to maximise breast development. In adulthood, oestradiol is typically given orally (at a dose of 1–2 mg) or transdermally (50 μg patch) as a maintenance dose with a cyclic progestin regimen to avoid endometrial hyperplasia [34]. These preparations were developed for the supplementation of sex steroids in post-menopausal women and therefore, dosages are lower compared to the endogenous oestrogen production in normal eugonadal women. It is unnecessary to administer cyclic progestins every month, and nowadays it is believed that less frequent withdrawal bleeding (e.g., every fourth month) is equally safe regarding the risk of endometrial cancer.

#### 2.3.2. Aims of Supplementing Gonadal Hormones in PWS

In PWS, supplementing gonadal hormones seems to be equally important not only because of the positive biological effects on muscle mass, strength, activity and bone density, but also regarding the psychosocial quality of life. Substitution has favourable effects on young men especially affected by the typical childish appearance with a lack of facial hair growth and voice break. For some girls and women, it is important to have menses. Oestrogen substitution also provides a concomitant contraceptive measure against unintended pregnancies that many parents wish to avoid.

Parents and caretakers want to see the lives of their children and adolescents progress smoothly and especially wish to avoid any associated drawbacks of increased aggressive behaviour as that seen in PWS boys and men. In girls, these drawbacks include any gain in fat mass and the hygienic aspects associated with menstruation.

#### 2.3.3. Outcome from Published Trials on Supplementing Gonadal Hormones in PWS

Supplementing gonadal hormones has only been studied in boys and men. Six boys with PWS were treated with hCG (Pregnyl^®^) for 2 years to induce puberty. The dosage regimen was dependent on bone age (e.g., 500 I.U. twice weekly for a bone age ranging from 13 to 13.5 years and 1000 I.U. twice weekly for a higher bone age of 13.6 to 14.5 years). This mode of treatment led to a rapid and sustained increase in testosterone levels (although still within the lower limit of normal), an adult pubic hair pattern and deepening of the voice. In one boy, complete testicular descent occurred during therapy. Testicular volume only slightly increased (max. 6 mL). FSH and LH levels decreased, and inhibin B remained below the normal range. It was concluded that fertility was not achieved during hCG therapy. Lean body mass increased and percentage fat mass did not decrease significantly during treatment. Behaviour was assessed and revealed pre-existing problems known in PWS: problems in social interaction, aggressiveness and irritability or rapid mood changes. These aspects did not change during therapy or were not markedly different from the pubertal changes observed in brothers and sisters [7,23].

Kido et al. [39] treated 22 adolescents and adults (aged 16 to 48 years) with hypogonadism (Tanner Stage less than 4 and testosterone level below 10.4 nmol/L) for 2 years with a relative low dose of 125 mg testosterone given intramuscularly per month. During treatment, the amount of pubic hair increased in 17 subjects, but characteristics on penile length and testicular volume were not documented. Erectile function and ejaculation were reported in eight and three subjects, respectively. Sperm content was negligible in semen samples collected from three subjects. Testosterone, LH and FSH levels remained unchanged throughout the 2-year study period. Body fat percentage decreased, and bone mineral density (BMD) and lean body mass increased. Behaviour was unchanged using the Modified Overt Aggression scale (MOAS). Behavioural problems (including shouting angrily, throwing objects around and skin picking) were already evident prior to beginning the study; testosterone treatment continued regardless of the observed behavioural problems.

In conclusion, the data on supplementing gonadal hormones in PWS are rather limited, confined to a single study for each age group, and behavioural problems did not increase during substitution therapy.

#### 2.3.4. Proposed Schedules for Supplementing Gonadal Hormones in PWS

While most children are treated nowadays with growth hormone under the supervision of a multidisciplinary team of health professionals, height, weight for height and body image have all improved. Based on this development, substitution of sex steroids has received great attention that has resulted in further proposals being made on how to advance the treatment of PWS hypogonadism in young subjects. Conversely, experience and recommendations for gonadal substitution in adults are sparse, most likely because most adults did not undergo this type of treatment in their childhood. We modified our search strategy described above to search specifically within PWS.

Most of the recommendations are based on expert opinion and the few studies reported in this paper. Heksch et al. [40] proposed beginning treatment for boys with a dose of 50–100 mg testosterone intramuscularly from the age of 14 years, if there is evidence of pubertal arrest. The initial dose can be gradually increased until the “typical” adult male dose of 250 mg/4 weeks is reached. For girls, hormone supplementation begins when there is no breast development by 13 years, pubertal progress stops, or menarche has not occurred by 16 years. Dose recommendations are not given, except that oral oestrogens should be administered in graduated doses with concomitant oral contraception after the onset of the first menstrual bleed. Hirsch et al. [6] proposed testosterone substitution in boys from 15–16 years. The working group of Bakker [41] recommend sex hormone supplementation from an age of 11 years in girls and 14 years in boys, unless there is normal progression of puberty; this is thought to prevent the decline in BMD, which they observed in PWS children treated with growth hormone hGH after the pubertal age.

At our institution, we administer testosterone at a mean age of 13.5 (12.5–15.5) years, when pubertal arrest becomes obvious in boys. Details are provided in Table 1. From a group of 27 boys and men, we delayed or discontinued treatment in seven subjects because of extreme (yet pre-existent) behavioural problems.

In girls, we recommend the induction of puberty when development stops. If puberty induction is necessary in girls, we begin at the age of 11.5 years. Details on the doses used are provided in Table 1. If puberty progresses, we most often start at the age of 14.5 years, usually when the girl expresses their desire to menstruate. The mean age was 13.3 (10–17) years in a total of 28 girls and women, where 22 underwent sex steroid supplementation at our clinic. Half were treated because of pubertal arrest and the remainder wanted to have menses. Half of the entire group stopped oestrogen supplementation because they no longer wanted to continue their menstrual cycle. We agreed with this strategy because BMD was normal (U.E., personal communication, December 2019).

For sex steroid-naïve adults, Kido et al. [39] recommends supplementation with monthly 125 mg testosterone injections, which are considered useful, safe and do not negatively affect behaviour.

### 2.4. Discussion and Prospects

From the current literature, it is clear that hypogonadism is frequent in boys and girls with PWS. Spontaneous puberty occurs more often in girls than boys and fertility and pregnancies have been reported in women only, in contrast, as a rule, men are infertile. The cause of the hypogonadism is mixed with a preponderance of primary hypogonadism, although varying combinations of primary and central hypogonadism are seen. PWS is a rare condition, therefore it is not surprising that controlled studies on the diagnosis and treatment of hypogonadism are absent. Observational studies, expert opinions and experience from treating patients with other conditions provide the evidence that forms a basis for our practice. From a biological point of view, supplementation of sex steroids is the way to proceed in improving body composition and bone mass as well as accomplish sexual maturation and develop as normal adolescents and people. Therefore, as a general rule, all PWS patients with hypogonadism should receive supplementation with sex steroids to induce puberty in children and adolescents and maintain normal physical development and health in adults when hypogonadism is diagnosed. In our opinion, sex steroid-naïve adults should also be offered supplements.

Nowadays, growth hormone treatment begins at a fairly young age, and catch-up growth is usually accomplished by most children when they reach the pubertal age. This suggests that final height will not be compromised by the induction of puberty at a normal age. Therefore, from a biological point of view, induction of puberty could take place from the age of 11 years in girls and 13 years in boys. When we consider the status of intellectual disability and delayed psychological maturation, one might consider inducing at a later age. However, the question arises as to when children and adolescents with PWS will achieve the appropriate age of maturity to deal with the effects of inducing puberty. Experience drawn from conditions such as Turner syndrome reveals good acceptance of the onset of (induced) puberty. Overall, we advocate the induction of puberty at the normal pubertal age, while taking into consideration that the starting age could be individualized based upon biological and psychological factors for children with PWS.

The main reason for the induction of puberty is to avoid the deceleration of growth and bone maturation. The observed deceleration in height velocity can provide a reliable clinical sign of hypogonadism particularly in boys, which can be ascertained by the absence of pubertal onset or progress and measurement of serum gonadotrophin and sex steroid levels. Further reasons for initiating and maintaining sex steroid supplementation are to achieve complete sexual maturation, an appropriate muscle mass and normal peak bone mass. In our experience, a normal appearance as an adult is especially very important for young adults with PWS. Data on the evolution of lean body mass and fat mass during growth hormone treatment show a body composition with a lower fat mass and higher lean body mass compared to that without treatment [4,5,42,43]. We have not found any published data on the effects of pubertal induction and sex steroid supplementation on the evolution of lean body mass and muscle mass after beginning sex steroid supplementation. In our own study, we could not demonstrate an increase in lean body mass after pubertal induction in boys with PWS. However, the effect on muscle mass is depending on physical activity, which was not examined, and long-term data are missing [23]. In terms of bone mass and BMD, a longitudinal study in children with PWS reported a decrease in BMD during growth hormone therapy after pubertal age when sex steroids were not supplemented [41]. In a 2-year observational study including young adults with PWS, Donze et al. showed an increase in BMD in young adults on sex steroid supplementation versus a decrease in those without sex steroid supplementation [44]. Therefore, supplementation of sex steroids could be important in normalising and maintaining BMD and peak bone mass. Whether this proposition has clinical relevance is not clear, although there is speculation on an increased fracture risk in adults with PWS [45,46]. In hypogonadal adults, the negative effects of withholding sex steroid supplementation on BMD and fracture risk are obvious, apparently through a combined effect of the absence of testosterone and oestrogens on bone metabolism [47].

The arguments against induction of puberty or starting sex steroid supplementation are also based on biological and psychological (i.e., behavioural) factors. Biological aspects encompass the event of spontaneous puberty or age-appropriate serum levels of sex steroids. Relative contraindications are previous thromboembolism, severe hypertension or a familial history of breast cancer. Psychological contraindications include active psychosis and behavioural problems, the latter being a relative contraindication. The biological contraindications are comparable to those outlined for sex steroid supplementation in post-menopausal women and Turner syndrome or the use of oral contraceptive pills in normal women. In PWS, the levels of sex steroids are normalised by supplementation and consequently, the pre-existent elevated risk for thromboembolic events in PWS [48] and the pre-existent risk for breast cancer are not increased. Furthermore, hormone replacement therapy (HRT) with oestradiol is commonly used in PWS instead of alkyl oestradiol, the active ingredient in contraceptive pills that have a different safety profile. Oestradiol has a beneficial effect on cardiovascular function (e.g., hypertension), and epidemiological and basic science studies do not support the commonly held assumption that HRT increases the risk of thrombosis [49]. The psychological and psychosocial considerations are specific for PWS, and behavioural problems are part of the PWS phenotype. We would like to raise two points on this subject: Firstly, behavioural problems seem to remain constant throughout childhood and the adolescent years. The few reports on induction of puberty and sex steroid supplementation do not present any increase in behavioural problems. We have also seen that there is no relevant change in behavioural problems at the start of or during sex steroid supplementation. In some adults, we observed the disappearance of mood swings after administering sex steroids. Secondly, there is a development in all PWS adolescents towards increased autonomy that can result in more conflict situations with parents and caretakers. Yet these events are also prevalent during normal pubertal development. This situation can be mistaken for aggressive behaviour in PWS subjects who often have less verbal and intellectual abilities to express their feelings and opinions. Parents and guardians, but also physicians, are often reluctant to start sex steroid treatment because the view is still held that testosterone leads to aggressive behaviour, although this is a fallacy. We generally consider biological contraindications as comparable to those encountered in normal girls and women, which should not affect supplementation of sex steroids in PWS. Behavioural problems should not be regarded as contraindications for the treatment with sex steroids.

The dose for induction of puberty in boys and girls with PWS can follow the schemes used in normal boys with hypogonadism and girls with Turner syndrome, respectively. These doses can be increased gradually every year over the course of 2 to 3 years until the adult doses are reached (Table 1). We, among other clinicians, did not use the same adult dose of testosterone in men with PWS but used about 75% of the dose used in normal men with hypogonadism. There is no basis or evidence for this policy and from a biological point of view, the application of a lower dose is inconsistent. We therefore suggest prescribing the same adult starting dose of testosterone to men with PWS and personalising the maintenance dose as in normal men with hypogonadism. In some men with visceral obesity, it might be better to monitor free testosterone levels during treatment for tailoring treatment [36]. In adolescents and women who are potentially fertile (e.g., inhibin B levels above 20 pg/mL), contraceptive measures should be taken; we prescribe an oral contraceptive pill in these cases.

An important issue in the treatment process is to decide whether to start and if so, when is the timing optimal. We believe supplementation should be started when growth decelerates around the pubertal age (at 13–14 years in boys and at 11–12 years in girls) or pubertal progress stops. This is applicable for most boys and about half of the girls. Naturally, the age at treatment initiation should be individualized based on biological and psychological factors. Measurement of serum sex steroid and gonadotrophin levels can provide additional information. Most of the remaining children and adolescents will show signs of hypogonadism at a later stage, whereby girls mostly have amenorrhea or oligomenorrhea. Who should be involved in this discussion? We think the discussion should be initiated by the physician before pubertal age has been reached. Information on the purpose, side effects, timing and decision process regarding supplementation therapy should be provided through direct consultation with the patient and guardians that is confirmed with a written report. When the physician decides that the time has come to start sex steroid supplementation, they should discuss the matter with the parents to organise how the child should be involved and who is responsible for informing the patient. In fact, it is preferable for the child or adolescent to be involved in the discussion and be active in making the final decision on whether treatment will take place and when. Appropriate monitoring of the therapeutic effects should be gathered. In Table 2, we provide a proposal for the measurements to be considered before starting and during supplementation of sex steroids. Close communication between children and adolescents with their parents and caretakers is necessary to ensure the smooth initiation of treatment and that reassurance can be provided during conflict situations. Another important issue focuses on how we should deal with the issue of women and their choice to stop oestrogen supplementation and menstruation. Longitudinal data on the development of BMD are needed to better counsel these women regarding whether and in which form sex steroids should be supplemented.

This review clearly shows that the evidence on hypogonadism in PWS and its treatment is limited. There are several questions to be solved regarding the biological and psychosocial effects of sex steroid supplementation. The long-term treatment effects on muscle mass and peak bone mass have yet to be studied and are important in supporting any future decisions on extended supplementation of sex steroids in PWS. Until these answers are established, we suggest supplementing sex steroids for hypogonadism in PWS in a similar manner to that for “normal” hypogonadism to prevent osteoporosis, sarcopenia and its negative consequences. A reliable and specific instrument to assess and rate behaviour in children and people with PWS in a longitudinal setting is lacking. We propose the development of a simple behavioural scale that may enable us to study the effect of sex steroid supplementation on behaviour among centres of excellence around the world by gathering sufficient data to solve this important issue.

### 2.5. Conclusions

Hypogonadism is prevalent in PWS, especially in males, and should be appropriately treated. Most of the evidence on the diagnosis and treatment of this condition is primarily based on expert opinion. In principle, we suggest initiating substitution with sex hormones at a similar age and using similar dosage regimen as that for normal hypogonadal children and adolescents. There are only a few absolute contraindications for substitution. Concerns about promoting aggressive behaviour are based on anecdotal reports. Normal maturation and striving for autonomy, which takes place during puberty should not be confused with aggression. We provide our expert opinion on the diagnosis, treatment and monitoring of hypogonadism in PWS. Further studies on treatment effects on muscle mass, (peak) bone mass and behavioural changes are needed to develop the best possible strategy for the diagnosis and treatment of hypogonadism in PWS.

## Figures and Tables

**Table 1 ijms-22-02705-t001:** Outline of our institutional sex steroid supplementation regimen to treat PWS patients after establishing pubertal arrest and hypogonadism.

	Age (Years)	Dosage
Boys, testosterone enanthate	13–14.5	100 mg/4 weeks
	>14.5	250 mg/4 weeks
	>18	1000 mg/12 weeks
Girls, oestradiol	11–12	0.5 mg/day
	12–14	1.0 mg/day
	>14	Cyclic dosing of with E2 2 mg/day E2 for 12 days, followed by 2 mg E2 + norethisterone acetate 1 mg for the next 10 days and 1 mg E2 for the last 6 days

E2, oestrogen.

**Table 2 ijms-22-02705-t002:** Examination strategy for sex steroid supplementation in PWS patients at our institution.

		at Start	after 1 Year	after 2 Years	Long Term, Every 2 Years
Boys	Clinical	Ht, Wt, PubSt, behaviour ^1^	Ht, Wt, PubSt, behaviour ^1^	Ht, Wt, PubSt, behaviour ^1^	Ht, Wt, PubSt, behaviour ^1^
	Lab	LH, FSH, T, E2, inhibin B, Hb, Hct	T, E2, Hb, Hct	LH, FSH, T, E2, inhibin B, Hb, Hct	T, E2, Hb, Hct
	Bone age	+	+	+	+ ^2^
	DXA	+		+	+ ^3^
Girls	Clinical	Ht, Wt, PubSt, behaviour ^1^	Ht, Wt, PubSt, behaviour ^1^	Ht, Wt, PubSt, behaviour ^1^	Ht, Wt, PubSt, behaviour ^1^
	Lab	LH, FSH, T, E2, inhibin B, AMH, liver panel	LH, FSH, T, E2, inhibin B, liver panel	LH, FSH, T, E2, inhibin B, liver panel	LH, FSH, T, E2, inhibin B, AMH, liver panel
	Bone age	+	+	+	+ ^2^
	DXA	+		+	+ ^3^

^1^ Behavioural scale to be developed, simple instrument to objectively measure behaviour. ^2^ until final height. ^3^ from age = 30 years every 5 years. AMH, anti-Müllerian hormone; DXA, Dual-energy X-ray absorptiometry; E2, oestrogen; FSH, follicle stimulating hormone; Hb, haemoglobin; Hct, haematocrit; Ht, height; LH, luteinising hormone; PubSt, pubertal stage; T, testosterone; Wt, weight.

## Data Availability

Not applicable.

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
