# Peer review of "Prader–Willi Syndrome and Hypogonadism: A Review Article"

_ijms, 2021, doi:10.3390/ijms22052705_

Round 1
Reviewer 1 Report
This report is a comprehensive review on hypogonadism in PWS patients, based on proceedings and outcomes of the International PWS Conference held in November 2019. The overall review is well formulated and clearly takes the reader through all aspects of hypogonadism, infertility, hormone levels, biochemical changes and treatment options for male and female PWS patients. There is enough information to make the paper interesting to both basic and clinical researchers, as well as clinicians who are seeking additional guidance on hormone replacement treatments for their patients.
Minor Suggestion--provide very brief historical context for full name of syndrome Prader-Labhart-Willi Syndrome, versus that without Labhart.
Author Response
Point 1: This report is a comprehensive review on hypogonadism in PWS patients, based on proceedings and outcomes of the International PWS Conference held in November 2019. The overall review is well formulated and clearly takes the reader through all aspects of hypogonadism, infertility, hormone levels, biochemical changes and treatment options for male and female PWS patients. There is enough information to make the paper interesting to both basic and clinical researchers, as well as clinicians who are seeking additional guidance on hormone replacement treatments for their patients.
Response 1: Thank you for your positive statement.
Point 2: Minor Suggestion--provide very brief historical context for full name of syndrome Prader-Labhart-Willi Syndrome, versus that without Labhart.
Response 2: We appreciate your suggestion and added a section on the history of the naming of the syndrome in lines 29-36. The story was passed years ago directly to one of the authors (UE) by Professor Prader.
Reviewer 2 Report
This review manuscript is written well. However, this need to be revised a few points.
1) Authors mentioned in this review of the abstract that we outline the current *******
Finally, we highlight additional areas of interest related to this topic and make recommendations for future studies.
However this was not show methods of the review, so could you describe this more specific?
2) Please describe how to select the literature in the Methods. I don't know the details.
3) There are too many sentences. Also, there are only two TABLEs. It is difficult to read, so please devise it.
4) There is little information about Table. Please describe in more detail.
Author Response
Point 1: Authors mentioned in this review of the abstract that we outline the current *******
Finally, we highlight additional areas of interest related to this topic and make recommendations for future studies.
However this was not show methods of the review, so could you describe this more specific?
Response 1: This paper is a narrative review or review article, based upon the literature, the knowledge of the authors, the proceedings and outcomes of the International PWS Conference held in November 2019. As usual in this type of review we describe and appraise previous work but do not describe specific methods by which the reviewed studies were identified, selected, and evaluated. The additional areas of interest and the recommendations for future studies are also based upon the sources mentioned before. To emphasis this we changed the naming of the review to “review article” and where appropriate we added the methods i.e. literature search strategies into the respective results sections.
Changes made:
- Line 2: we changed the naming of the review from “PWS and Hypogonadism: A Review” to “PWS and Hypogonadism: A Review Article”
- Lines 16-17: In the Abstract we added a sentence to provide the sources for this review article The section now reads as: “…In this review, we outline the current knowledge on the clinical, biochemical, genetic and histological features of hypogonadism in PWS and its treatment. This was based on current literature and the proceedings and outcomes of the International PWS annual conference held in November 2019. We also present our expert opinion regarding the diagnosis…
- Lines 57-59: We extended one sentence of the Introduction, which now reads as: “Consequently, this paper aims to describe the current knowledge on hypogonadism in PWS children and adults with respect to diagnosis and treatment, based on current literature and the outcomes of this meeting. We present our expert opinion …”
- Lines 226-229: We added our search strategy at the beginning of the results section on 2.3.1 Supplementing gonadal hormones in normal male and female hypogonadism: The additional text reads as: “We searched PubMed with emphasis on publications from 2015 and later using the keywords: “hypogonadism”, “treatment”, “substitution” in combination with the “similar articles” and “cited by” tools to retrieve current treatment guidelines as basis for the next section.”
- Lines 344-345: We added our search strategy at the beginning of the results section on 2.3.4 Proposed schedules for supplementing gonadal hormones in PWS: The additional text reads as: “We modified our search strategy described above to search specifically within PWS.
Point 2: Please describe how to select the literature in the Methods. I don't know the details.
Response 2: see our answers above
Point 3: There are too many sentences. Also, there are only two TABLEs. It is difficult to read, so please devise it.
Response 3: We understand your objection. Our text is long. But from our point of view, that is the nature of a narrative review. We cannot identify any other text passage that would look tidier with a table. Therefore, we shortened the text of paragraph 2.2.1 line 94-95; line 98-100; line 103-104. We shortened the text of paragraph 2.2.2 considerably (current lines 116-119, 120-133, 135-159). In paragraph 2.2.3 for clarity we moved the first sentence to the end of the paragraph line 170-172 to lines 177-179 cm. In paragraph 2.2.4 we shortened line 191-193. In paragraph 2.3.1 we changed lines 255-257, line 268 and line 273. In paragraph 2.3.2 we shortened lines 294-296 and lines 298-299. In paragraph 2.3.3 we shortened and for clarity we changed lines 327-330. In paragraph 2.3.4 we shortened line 355, lines 369-371 and lines 373-379. In paragraph 2.4 (discussion and prospects) we shortened lines 415-417, lines 428-429; for clarity we changed 434-43 and we shortened lines 448-450, 529-531 and 546-547.
In detail:
- Line 420 and 430: We added textual references to Table 1.
- Lines 365-367: We moved Table 1 to a more appropriate location and added a legend for abbreviations
- Line 372-374: The former lines 330 to 335 were shortened as: “In girls, we recommend the induction of puberty when development stops. If puberty induction is necessary in girls, we begin at the age of 11.5 years. Details on the doses used are provided in Table 1.”
Point 4: There is little information about Table. Please describe in more detail.
Response 4: see our answers to point 3 / Table 1 above
Further corrections made:
Lines 18, 60….: “counseling” was corrected as “counselling”
Lines 133….: “hypogonadotrophic” was corrected as “hypogonadotropic”
Line 185: “hypergonadotrophic” was corrected as “hypergonadotropic”
Lines 484, 143-146, 151-152 …., minor spelling corrections or editing
Round 2
Reviewer 2 Report
Thanks for your revision based on the reviewer's comments. There is no more comments for this revised manuscript. So, I recommend to publish this asap.